# "I was given PrEP, but had no privacy": Mystery shopper perspectives of PrEP counseling for adolescent girls and young women in Kisumu County, Kenya

**Melissa Vera**[1¤*], **Helen Aketch**[2☉], **Caroline Omom**[2☉], **Felix Otieno**[2], **George Owiti**[2], **Joseph Sila**[2], **John Kinuthia**[2], **Kristin Beima-Sofie**[1], **Jillian Pintye**[1], **Valarie Kemunto**[2], **Eunita Akim**[2], **Grace John-Stewart**[1], **Pamela Kohler**[1]

1 School of Nursing/Global Health, University of Washington, Seattle, WA, United States of America,
2 Centre for Clinical Research, Kenya Medical Research Institute, Kisumu, Kenya

☉ These authors contributed equally to this work.
¤ Current address: College of Nursing, Washington State University, Spokane, WA, United States of America
* melissa.vera@wsu.edu

**Data Availability Statement:** Data cannot be shared publicly because of the population age (some are minors). Data are available from the

## Abstract

Pre-exposure prophylaxis (PrEP) is being scaled up to prevent HIV acquisition among adolescent girls and young women (AGYW) in Eastern and Southern Africa. In a prior study more than one-third of AGYW 'mystery shoppers' stated they would not return to care based on interactions with health providers. We examined the experiences of AGYW in this study to identify main barriers to effective PrEP services. Unannounced patient actors (USP/'mystery shoppers') posed as AGYWs seeking PrEP using standardized scenarios 8 months after providers had received training to improve PrEP services. We conducted targeted debriefings using open-ended questions to assess PrEP service provision and counseling quality with USPs immediately following their visit. Debriefings were audio-recorded and transcribed. Transcripts were analyzed using thematic analysis to explore why USPs reported either positive or negative encounters. We conducted 91 USP debriefings at 24 facilities and identified three primary influences on PrEP service experiences: 1) Privacy improved likelihood of continuing care, 2) respectful attitudes created a safe environment for USPs, and 3) patient-centered communication improved the experience and increased confidence for PrEP initiation among USPs. Privacy and provider attitudes were primary drivers that influenced decision-making around PrEP in USP debriefs. Access to privacy and improving provider attitudes is important for scale-up of PrEP to AGYW.

## Introduction

Adolescent girls and young women (AGYW) in Eastern and Southern Africa experience disproportionately high HIV incidence compared to other age groups [1]. Although, new HIV infections among young people ages 15–24 have declined over the past decade (56% decline for males and 42% for AGYW) [2], the incidence of new HIV infections are 3-fold higher

University of Washington Institutional Data Access / Ethics Committee (contact via email: hsdinfo@uw.edu, or phone: +1-206-543-0098) for researchers who meet the criteria for access to confidential data.

**Funding:** This study was funded through P.K. by the National Institutes of Health (NIH, R01 HD094630) and M.V. received additional funding by the National Institute of Nursing Research (NINR, T32NR019761). "The funders had no role in study design, data collection and analysis, decision to publish, or preparation of the manuscript."

**Competing interests:** The authors have declared that no competing interests exist.

among AGYW (32%) than their male peers (10%) in this region [3]. In Kenya, new HIV infections among AGYW are approximately twice that of their male peers annually (approximately 12,500 compared to 6,300, respectively) [4].

Pre-exposure prophylaxis (PrEP) is recommended by the World Health Organization (WHO) [5] and the Kenya Ministry of Health for those at-risk for acquiring HIV, with tenofovir (TFV)-based daily oral PrEP scaling up [6] in Kenya primarily through local hospitals, maternal child health (MCH) clinics, and family planning (FP) clinics to reach AGYW [7–9]. Despite these ongoing efforts to increase PrEP access among AGYW in Kenya, initiation in this priority population remains suboptimal, with studies suggesting only 4–16% of Kenyan AGYW with behaviors associated with HIV acquisition initiate PrEP when offered in FP clinics [7, 8, 10]. In one study, among those who initiated PrEP, only 37% of AGYW persisted with PrEP medication after three months [11].

Negative health care user experiences and stigmatizing interactions with health providers contribute to low initiation of PrEP among AGYW by dissuading their engagement in care [9, 12–16]. Stigma has been shown to have devastating consequences to health outcomes in the HIV care continuum, including loss of social support, isolation, depression, and decreased utilization of preventive care [17]. Accessing and maintaining care is key to correctly taking PrEP and ultimately preventing HIV acquisition [18]. Thus, stigmatizing interactions with health care providers are a key barrier to address in order to scale-up PrEP in AGYW.

During a cluster randomized trial of a clinical training intervention to improve quality of provider-patient PrEP interactions through destigmatization [19], we used unannounced standardized patient actors (USPs) [20] posing as AGYW seeking PrEP at health facilities in Kenya [21]. During the baseline phase of this study, USPs quantitively measured quality of PrEP counseling delivery. Overall, 36.5% of USPs stated they would not want to return to the health care provider they saw due to stigmatizing interactions [22]. To explain USP ratings, and gain a more in-depth understanding of USP experiences, we conducted a qualitative analysis of debriefing interviews following the unannounced encounters.

## Materials and methods

### Study design

We conducted a qualitative analysis of audio-recorded debriefings between trained USPs and study staff following visits by actors posing as AGYW seeking PrEP services.

### Population and setting

During a randomized clinical trial, USPs presented to 24 facilities (12 intervention, 12 control) to assess quality of PrEP services. Facilities for the trial were purposively selected based on expected AGYW patient volume, and included FP and MCH facilities from the county, sub-county, and health center levels. Facilities were an even mix of urban, peri-urban, and rural settings, including public and private/faith-based facilities. Health providers at intervention sites participated in an educational training intervention. All health providers at study facilities were eligible to participate in USP visits, as long as they were 18 years or older, provided PrEP services to AGYW, currently employed at the study facilities, and able to provide informed consent).

### Data collection

Eight months post-training, eight professional, Kenyan actors (aged 19–23 years) visited facilities posed as PrEP seeking AGYW. Two USPs were assigned to each case (some overlapping

**Table 1. PrEP scenarios depicted by patient actors.**

| Case Number | Case Type | Scenario Depicted |
|---|---|---|
| 1 | New PrEP initiator | 18-year-old female seeking information about PrEP; husband's HIV status is unknown, he does not use condoms. |
| 2 | PrEP continuation, adherence challenge | 24-year-old female seeking PrEP refill; her husband is HIV-positive and on ART. She has an irregular daily schedule, making it harder for her to adhere to daily PrEP. |
| 3 | PrEP continuation, transfer-of-care case | 19-year-old female transferring PrEP care to a new facility; engaging in transactional sex relationships with multiple partners including one main older "sponsor"; HIV status is unknown for all partners and they do not use condoms. |
| 4 | PrEP continuation, adherence challenge | 17-year-old female seeking PrEP refill; her boyfriend is a truck driver and is out of town for long periods of time; she is adherent to daily PrEP when he is in town, but stops when he is away. |
| 5 | Seeking contraception | 16-year-old female seeking contraception; her boyfriend's HIV status is unknown. |
| 6 | New PrEP initiator, young AGYW | 18-year-old female seeking information about PrEP; her boyfriend's HIV status is unknown and they do not use condoms regularly. Now that she is sexually active, the girl is seeking self-controlled HIV prevention. |

cases) using scripted commonly occurring PrEP scenarios (Table 1) to seek PrEP services at all 24 facilities. Immediately following USP encounters, research assistants trained in qualitative methods (E.A. and V.K.) guided the USPs through a debriefing process. Debriefings included a quantitative checklist to capture primary trial endpoints, and open-ended questions to explore why USPs rated the encounters either positively or negatively (Table 2). Research assistants were provided with a list of probing questions to elicit sufficient details and concepts to explore why the USP rated the provider as they did and how the experience felt for them. Debriefings took place in local languages, if necessary, were audio-recorded, and covered themes related to provider language use, stigmatizing or judgmental behaviors, time allowed for questions, respectful or disrespectful behaviors, privacy, listening skills, and whether the USP would return to the provider in the future. USP debriefs were transcribed verbatim by one Kenyan study staff (C.K.), and initial transcripts were independently verified against the original audio file by a member of the analysis team (M.V.). Repeat interviews and review of transcripts for correction were not offered to SP participants due to COVID-19-related travel restrictions and logistical challenges.

**Table 2. Quantitative checklist measurement objectives with qualitative additions for end-line debriefing.**

| Quantitative Measurement Objective | Qualitative Question Added |
|---|---|
| Language use | In what ways did the provider use language that was easy or hard to understand? |
| Judgment/stigma | Were there specific things the provider said or did that made you feel judged, if at all? |
| AGYW questions during encounter | What did the provider say or do to make you feel encouraged or discouraged to ask questions? |
| Respect | What were specific things the provider did to make you feel respected or disrespected? |
| Privacy | What specifically made you feel your privacy was or was not being protected? |
| Listening skills | What specific things did the provider say or do that showed that he/she was listening or not? |
| Return to provider | Would you go back to see this provider? Why or why not? |

## Data analysis

Transcripts were uploaded into Dedoose version 9.0.54 (SocioCultural Research Consultants, LLC, Los Angeles, CA, USA). Using principles of thematic analysis [23, 24], the analysis team (M.V., C.K., H.A., and C.O.) reviewed transcripts to code, identify, and evaluate themes related to *positive user experience* concepts from the High-Quality Health System Framework [25]. The High-Quality Health System Framework focuses on health system function, user experience, and how people benefit from healthcare. *Positive user experience* encompasses two concepts: respect and user focus. *Respect* incorporates dignity, privacy, non-discrimination, autonomy, confidentiality, and clear communication. *User focus* incorporates choice of provider, short wait times, patient voice and values, affordability, and ease of use. The analysis team developed a codebook from a close read of a subset of 15 transcripts using inductive and deductive methods. Deductive codes were derived from concepts of *positive user experience* in the High-Quality Health System Framework. Inductive codes were obtained through multiple reviews of the transcripts by the analysis team. The codebook was iteratively developed, with the clarification of codes and code definitions occurring through discussion and consensus as more transcripts were reviewed. After the codebook was finalized, transcripts were coded independently by a member of the analysis team and received a secondary review by a different analysis team member. Differences in coding were discussed until resolved through consensus. Coded data were put into framework matrices to identify themes across all 91 debriefings [26].

## Ethical considerations

Participant HCWs provided written informed consent and knew they would be visited by a USP but were not given details of the time or date of the visit. HCW names were not linked with assessment data. Data analyses were carried out in Kisumu County, Kenya and Seattle, Washington. The University of Washington Institutional Review Board (IRB, approval number CR00006099) and the Kenyatta National Hospital Ethical Review Committee (ERC, approval number P751/10/2018) reviewed and approved the conduct of this study.

## Results and discussion

In total, 91 USP debriefings were audio-recorded and transcribed for analyses: 18 (20%) depicted case 1, 14 (15%) case 2, 15 (16%) case 3, 14 (15%) case 4, 17 (19%) case 5, 13 (14%) case 6 (Table 2). Overall, USPs reported a mix of positive and negative experiences receiving care from providers, where an encounter that incorporated privacy, compassionate communication, and education on PrEP elicited a more favorable experience for the USP. We identified three major themes related to PrEP seeking experiences from the concept of *respect* under the domain of *positive user experience* from the High-Quality Health System Framework[25]: 1) privacy improved willingness to continue in care, 2) respectful attitudes created a safe and caring experience for USPs, and 3) patient-centered communication improved USP experience and PrEP initiation confidence.

## Privacy as a determining factor for returning to a provider

Most USPs reported that privacy was of great concern for them while visiting a healthcare facility for PrEP. However, multiple USPs described having several people in the room during their encounter with the provider. Interruptions by other patients or providers made some feel they were not in a safe space to receive PrEP counseling. Some USPs felt that the amount of people and open doors, paired with loud provider voices, created feelings of unease and limited sharing honest information with their provider.

"*The door was wide open, in the room there were four other people, one other person came in during our encounter, and he consulted somebody else concerning the records. When we were in the encounter, he would be asking me questions and also speaking with other colleagues in the room, so it wasn't private. Then there was another client who came in [and] just interrupted our session, so I did not feel like my privacy was protected.*" (case 1, married new PrEP initiator)

Lack of privacy was a common concern, and multiple USPs responded positively about all aspects of their encounter with their provider except the privacy aspect. These USPs reported unwillingness to return because of this lack of privacy, even though the encounter was generally favorable.

"*I would go back to the medical provider because. . .she respected me and treated me without being judgmental and she also offered me the medication. That will make me go back to the healthcare provider, but I would probably not go back because of the privacy state. There were so many people in the room and the interruptions, so I felt that my privacy was not 100% guaranteed.*" (case 3, transactional sex)

In contrast, USPs who described situations where their privacy was protected stated that they would return to the same provider for care because they felt respected. Providers who were aware of their surroundings and shifted the direction and questions within the encounter accordingly, were viewed positively.

"*It was just the two of us in the room and even during the encounter when somebody else called her from outside she told them to wait a little bit because she was in the session with me, so I felt. . .that my privacy was protected, and she was concerned about handling me without having interruptions.*" (case 5, young seeking contraception)

## Respectful attitudes created a safe and caring experience for USPs

Provider attitude was an important aspect in deciding whether one would want to return to the same provider. USPs described providers with good attitudes as those who were friendly, personable, and non-judgmental, showed they cared about the patient by asking about their personal lives. Providers with good attitudes were also described as being vulnerable, honest, and thorough, offering help and providing additional resources.

"*I will go back to the healthcare provider because. . .she was very friendly. She inquired how I was doing, my personal health, [my] family. . .She apologized for keeping me [and] was respectful by. . .address[ing] me by my name. She even apologized for not having condoms, but then she advised me to go to a public hospital where I could be given condoms. She was not judgmental. . .She was just nice (both laugh).*" (case 1, married new PrEP initiator)

On the other hand, providers who were confrontational were viewed as having a negative attitude. One USP described a confrontational provider who used a harsh tone of voice, was terse in communication style, fidgeted in the chair, looked through files, and did not maintain eye contact or adopt an open body posture. In addition to being viewed as confrontational, the USP described this provider as being disorganized, uncaring, and unprofessional, which came across as disrespectful to the USP.

"*When I entered the room, the first thing he asked me was, 'Why are you here?'...I told him I am here for PrEP and then he told me, 'That is not a reason.'...he was fidgeting [in] the chair and there were some things he was noting down...he was doing that while one leg was on top of the chair and then...he stopped attending to me and he started looking for some other files, sitting on the table, so I felt disrespected.*" (case 6, new PrEP initiator)

Judgment played an important role in how provider attitudes were viewed by USPs. Providers that were non-judgmental were often aligned to those having a positive attitude and encouraged USPs to continue care with them by being attentive, active listeners.

"*She was attentive, she was really listening to me and she never judged me from my situation [because] I told her that I have a partner who has a wife and she was never judgmental... the body posture and the language showed respect, so I felt respected.*" (case 2, serodiscordant adherence challenge)

Among USPs who described feeling judged, evidence of judgment came through verbal language, posture, tone, and body language. Judgmental providers not only dissuaded USPs from continuing care, but they limited USP willingness to be open and honest about their care and questions. USPs also viewed unwillingness or reluctance to provide PrEP as being judgmental.

"*The clinician said I was not eligible for PrEP because he felt that only sex workers and miscoded couples should be given PrEP...He told me that most people waste drugs because they don't take them, so he said that I should go and think twice.*" (case 6, new PrEP initiator)

Some USPs experienced judgment not only from providers, but from staff in the pharmacy or reception areas. Similarly, these judgmental encounters made them feel very uncomfortable and less likely to seek further care, as this negative treatment experience occurred in spaces that were more public than the provider's examination rooms.

"*In the pharmacy, the nurse was very judgmental, and she was asking me why I am using PrEP [because I] am still young; that I should just dump my boyfriend because it is not worth sacrificing my life. She even sneered.*" (case 6, new PrEP initiator)

Body language was seen to impact how USPs felt during their encounter. Some reported that the provider seemed closed-off in demeanor, did not face them directly, did not employ much eye contact, with one USP noting that their provider was physically handling papers and files not related to the USP during their encounter.

"*Most of the time he was not concerned with whatever I was saying. He was just concentrating on writing and doing other things in the office, in the file, which I didn't know what he was doing. So, I did not feel respected, also because he did not maintain eye contact at all. Even [while] asking me questions about the session or the drug, he was just doing his own things and ticking the file, so I didn't feel respected at all. So, I will not go back.*" (case 6, new PrEP initiator)

Conversely, providers that made eye contact, nodded that they were listening and showed through their body positioning that they were paying attention, were seen as creating a welcoming environment where the USP could have issues addressed, and further, that the provider cared about the USP's health.

"*He was very welcoming, his tone was soft and friendly. . .During the whole session he was attentive. He would maintain eye contact with me. He was listening by his body language. . .nodding the head. [When I] said I had a problem forgetting to swallow my drugs sometimes, he turned now facing me directly. . .so that he could find out why. . .I forget [and] what can we do to sort out the issue.*" (case 3, transactional sex)

## Patient-centered communication improved USP experience and PrEP initiation confidence

USPs reported several aspects of patient-centered communication that influenced their feelings about their care experiences. Showing care through listening made some USPs want to come back for further care. Providers that were seen as "interactive" were viewed more positively by USPs. Being interactive included asking questions about the USP and asking if the USP had questions for the provider.

"*At. . .first, she counselled and gave an opportunity to ask any questions if I had any and then continued with the counseling. Then she also gave me an opportunity where she told me that she is not going to talk anymore unless I ask a question.*" (case 1, married new PrEP initiator)

USPs that did not receive an interactive session were less willing to return to the provider for further care because they didn't feel their questions would be answered, which decreased their confidence in taking PrEP. Allowing more time to interact with the provider was seen as giving better care. USPs who had providers that spent plenty of time with them, asking questions and providing guidance, reported feeling cared for and understood.

"*I felt that she was very welcoming and understanding. . .She took me through an elaborate counseling session on how I am supposed to take my drugs and how it would affect me, not only because I'll get HIV, but how. . .I would develop resistance to the drugs, and they wouldn't be of help to me. And also when she was describing how. . .I should speak to my partner so that we can come for testing, she even went further to say if I found that my partner is positive, they would offer counseling on how we can take the next step. . .I found that she treated me with a lot of care and understanding.*" (case 3, transactional sex)

USPs who did not feel they received quality time with or the opportunity to ask questions of their provider were hesitant to return for follow-up care, even if PrEP was prescribed.

"*The provider dominated the whole encounter, there wasn't any point where I was given a chance to ask any question or to raise any concern it was just him talking through the whole encounter and me answering any questions that he asked.*" (case 3, transactional sex)

USPs noted how verbal and non-verbal language (i.e., "body language") used by providers were important aspects of receiving patient-centered care. Using words that were familiar (i.e., not medical jargon), or were accompanied by explanations if unfamiliar, was seen positively by most USPs. Simple, easy to follow language allowed USPs to understand more fully what the provider was communicating, increasing their positive regard for the encounter.

"*Throughout the session, from the start to the end, she used a simple language that I could understand. . .There is a point where she used a medical term adherence, she went ahead and explained [that] adherence means taking the medication. . .daily, consistently, the way [we]*"

*had been told by the physician. . .so I understood [what] she said." (case 1, married new PrEP initiator)*

This analysis complemented and explained quantitative endpoints of a clustered randomized trial assessing a clinical training intervention for healthcare providers [22]. We found that AGYW USPs preferred to return to providers who protected their privacy, didn't allow for interruptions to happen during the encounter, maintained respectful and caring verbal and non-verbal communication, and allowed time for questions and concerns to be addressed. This adds to the broader context in which AGYWs are seeking HIV prevention services where stigma reduction is paramount and changing discriminating attitudes towards HIV prevention and sex are crucial to decreasing HIV transmission. If AGYW can receive better care in the setting they receive care most often (i.e. within the healthcare system among health care providers), it could help with HIV prevention efforts among this population.

Privacy was a priority for USPs in our study. A study among AGYW in South Africa similarly found that privacy was of great concern among AGYW and hindered them from accessing sexual and reproductive health care due to fears of stigmatizing behaviors from clinical and nonclinical staff, to being "outed" by structural issues that allow AGYW to be physically seen by community members in clinics while interacting with a provider [27]. Similarly, one study among young, Black men who have sex with men (YBMSM) in the US found that fear of being physically seen by community members (especially family, friends, and church community members) deterred YBMSM from accessing HIV testing [28]. Another study in Kenya found that improving access to PrEP (including HIV testing, counseling, clinical assessment, and drug dispensing) in a one-stop-shop (OSS) model improved privacy issues by decreasing the amount of movement from place to place to receive all the services needed for HIV prevention care [29]. Our study suggests that measures to enhance patient privacy by improving the physical space in clinics may be a key aspect to improving PrEP delivery, as it was reported as a barrier by USPs even when the provider's communication skills were acceptable. Further, our study suggests improving provider and nonclinical staff behaviors are essential steps toward improving AGYW experiences seeking PrEP services and can potentially decrease fears of anticipated stigma.

Connected to and in addition to privacy, respectful attitudes among healthcare providers were found in our study to be essential for creating a safe and positive experience for AGYW in the clinic setting. Similar findings have been described in other settings, where disrespectful or stigmatizing interactions with health providers are consistently reported as a barrier to continuing care across a number of populations and health conditions [30]. The resulting anticipated stigma also has been shown to affect health outcomes, including delays in treatment [30], life-threatening disease complications [31], and not accessing healthcare [27]. Other research indicates that providers may not be aware of the stigma they are inflicting on AGYW and that their own beliefs and values influence willingness to offer PrEP to young people. In South Africa, investigators found that 75% of clinical and nonclinical staff felt providing PrEP to AGYW would increase behaviors of sexual risk, pregnancy, and STI incidence [27]. Our study contributes to the global narrative by highlighting the unique stigma faced by AGYW seeking PrEP and offering more perspectives that providers need to augment or adjust their communication skills to decrease stigmatizing behaviors, even if they believe they are not being stigmatizing.

Finally, patient-centered communication was the last theme found in our study that can affect AGYW's confidence in PrEP and overall experience seeking care. The Lancet Global Health Commission on High-Quality Health Systems defines respectful care as having interactions between providers and patients that incorporate dignity, privacy, non-discrimination,

autonomy, confidentiality, and uses clear communication [25]. Patient-centered communication is part of a high-quality health system and can improve patient outcomes and how they feel about continuing care with a provider [32]. Patient-centered communication should elicit the patient's concerns, values, expectations for care, available resources, and informed consent for their care plan. Patient-centered communication ideally incorporates relationship building, sensitivity to patient needs, compassion, empowering discussion of care, and respect for patient privacy and decisions. The findings of our study suggest that implementing better patient-centered communication during encounters between providers and AGYWs seeking PrEP could potentially increase PrEP initiation and continuation of care among this population.

In the broader context of HIV prevention among AGYW, more is needed: more access points to PrEP, better systems for retention of AGYW to PrEP services, an overall cultural shift toward more understanding and acceptance of AGYW sexual practices and what they do to protect themselves. Retail pharmacies are a promising new addition to PrEP acquisition [33]. Purchasing PrEP over-the-counter has shown in recent studies to provide the privacy, good communication, and convenience many AGYWs prefer when accessing PrEP. Peer-support groups are another way to potentially increase PrEP intiation among AGYWs [34]. This model utilizes experienced AGYWs to help PrEP-naïve AGYWs take HIV self-tests and then refers them to PrEP services at clinic. Peer-supported PrEP initiation may be another way to bolster PrEP initiation among AGYWs, but they still need a good experience at the clinic once they get there. Our study reinforces the potential impact that health care providers have among AGYWs accessing HIV prevention.

## Limitations

Our study draws upon data from standardized patient actors portraying young women seeking PrEP and may not be generalizable to the greater AGYW communities of Kenya or other HIV high-burden countries globally. AGYW in the community who were not recruited may have different challenges that we are unable to include in these analyses. To make a qualitative analysis seamless we added open-ended questions to an already-existing quantitative checklist, which may have influenced what actors perceived as good care. Lastly, we completed debriefings from encounters with providers who were involved in an educational intervention in the parent study, possibly skewing how USPs experienced care. This may not be generalizable to the greater community of PrEP providers.

## Conclusions

Privacy and provider attitudes were primary drivers of return to care decision-making among actors portraying AGYW at Kenyan clinics. Improving AGYW experiences in the clinic setting is critical to improving PrEP service delivery. Future research is needed that includes a broader sample of non-actor AGYW participants to evaluate whether provider and health systems interventions to improve these interactions will result in improved uptake and persistence in this priority population. Further, more research is needed to better contextualize and understand the benefits of retail pharmacy-based PrEP services and peer-supported PrEP programs, both aimed at increasing PrEP access points and retaining more clients on PrEP for younger populations.

## Supporting information

**S1 File.**
(DOCX)

## Acknowledgments

We would like to acknowledge the AGYW participants who gave their trust to us to perform the interviews, our study staff who worked tirelessly to implement this work.

## Author Contributions

**Conceptualization:** Felix Otieno, George Owiti, Joseph Sila, John Kinuthia, Jillian Pintye, Grace John-Stewart, Pamela Kohler.

**Data curation:** Helen Aketch, Caroline Omom, Felix Otieno, George Owiti, Joseph Sila, Valarie Kemunto, Eunita Akim.

**Formal analysis:** Melissa Vera, Helen Aketch, Caroline Omom, Kristin Beima-Sofie.

**Funding acquisition:** Melissa Vera, Jillian Pintye, Grace John-Stewart, Pamela Kohler.

**Investigation:** Valarie Kemunto, Eunita Akim, Grace John-Stewart.

**Methodology:** Joseph Sila, Kristin Beima-Sofie, Jillian Pintye.

**Project administration:** Felix Otieno, George Owiti, Joseph Sila, Jillian Pintye, Pamela Kohler.

**Resources:** Felix Otieno, George Owiti, Kristin Beima-Sofie.

**Software:** Joseph Sila.

**Supervision:** Felix Otieno, John Kinuthia, Kristin Beima-Sofie, Jillian Pintye, Grace John-Stewart, Pamela Kohler.

**Writing – original draft:** Melissa Vera.

**Writing – review & editing:** Melissa Vera, Helen Aketch, Caroline Omom, Felix Otieno, George Owiti, Joseph Sila, John Kinuthia, Kristin Beima-Sofie, Jillian Pintye, Valarie Kemunto, Eunita Akim, Grace John-Stewart, Pamela Kohler.

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
