## [Decision Letter · Decision Letter 0]

5 Apr 2024

PONE-D-24-01290“I was given PrEP, but had no privacy”: Mystery shopper perspectives of PrEP counseling for adolescent girls and young women in Kisumu County, KenyaPLOS ONE

Dear Dr. Vera,

Thank you for submitting your manuscript to PLOS ONE. After careful consideration, we feel that it has merit but does not fully meet PLOS ONE’s publication criteria as it currently stands. Therefore, we invite you to submit a revised version of the manuscript that addresses the points raised during the review process.

We look forward to receiving your revised manuscript.

Kind regards,

Matt A Price

Academic Editor

PLOS ONE

Journal Requirements:

This study was funded through P.K. by the National Institutes of Health (NIH, R01 HD094630) and M.V. received additional funding by the National Institute of Nursing Research (NINR, T32NR019761).

4. In the online submission form, you indicated that your data is available only on request from a third party. Please note that your Data Availability Statement is currently missing [the contact details for the third party, such as an email address or a link to where data requests can be made]. Please update your statement with the missing information. 

Additional Editor Comments:

I have reviewed and have only minor comments. In general, the paper is well written and easy to follow. Two minor suggestions:

Results, first sentence: what is “case 1”, “case 2” etc. I thought there were 8 actors/USPs not 6, is this the number of interviews from each USP? I see from the table this refers to scenarios or scripts. Perhaps this could be better described in the methods?

Did each USP participate in each scenario/case, or did some “specialize” in certain cases, so to speak?

Reviewers' comments:

Reviewer's Responses to Questions

**Comments to the Author**

1. Is the manuscript technically sound, and do the data support the conclusions?

Reviewer #1: Yes

2. Has the statistical analysis been performed appropriately and rigorously? 

Reviewer #1: N/A

3. Have the authors made all data underlying the findings in their manuscript fully available?

Reviewer #1: Yes

4. Is the manuscript presented in an intelligible fashion and written in standard English?

Reviewer #1: Yes

5. Review Comments to the Author

**Reviewer #1:** Authors present a large qualitative study of mystery clients who presented as AGYW who were interested to discuss PrEP continuation/prescription. The study was embedded in a large trial involving 24 health facilities. Data were thematically analysed and revealed three themes reflecting on the interactions experienced.

Privacy, attitudes to clients, and 'patient-centered communication' - emerge as main perceived determinants of improved HIV preventive care (i.e. PrEP continuation and initiation).

The paper is well-written reinforces that clinician attitudes impact uptake of prevention products.

The discussion is fairly brief and reiterates the three major findings. I was missing a broader discussion on what it will take to get AGYW to take up daily oral PrEP.

It is clear that improved privacy and client-centred communication is one aspect. There are several other aspects of PrEP programming (e.g. stigma reduction, societal attitudes to sexual activity and PrEP, mental health support, alternative PrEP dispensing approach, role of peer support) that go beyond the mystery clients approach but could be mentioned to contextualise study findings.

The final sentence of the discussion is weak: "Future research is needed that includes a broader

sample of non-actor AGYW participants to evaluate of whether provider and health systems interventions to improve these interactions will result in improved uptake and persistence in this priority population."

Especially since authors speak about health systems interventions that have hardly been discussed or put in context.

Small point: how did the sex (gender) of the provider impact interactions?

6. PLOS authors have the option to publish the peer review history of their article (what does this mean?). If published, this will include your full peer review and any attached files.

Reviewer #1: **Yes: **Eduard Sanders

---

## [Author Response · Author response to Decision Letter 0]

3 Jul 2024

Response to Reviewers

Dear Matt Price and Eduard Sanders,

Thank you for your feedback and time. Below are my responses to your edits for Manuscript: PONE-D-24-01290, “I was given PrEP, but had no privacy”: Mystery shopper perspectives of PrEP counseling for adolescent girls and young women in Kisumu County, Kenya.

Journal Requirements:

Addressed. I reviewed the PDF links and reformatted the title page and main manuscript per PLOS One requirements.

2. Please include a complete copy of PLOS’ questionnaire on inclusivity in global research in your revised manuscript.

Completed and uploaded to submission.

This study was funded through P.K. by the National Institutes of Health (NIH, R01 HD094630) and M.V. received additional funding by the National Institute of Nursing Research (NINR, T32NR019761).

Completed and changed in the Cover Letter. Added: "The funders had no role in study design, data collection and analysis, decision to publish, or preparation of the manuscript."

4. In the online submission form, you indicated that your data is available only on request from a third party. Please note that your Data Availability Statement is currently missing [the contact details for the third party, such as an email address or a link to where data requests can be made]. Please update your statement with the missing information.

Amended the online statement with an email and phone number to contact the third party (UW Human Subjects Research Dept.).

Reviewed References and did not need to retract any citations.

Additional Editor Comments:

I have reviewed and have only minor comments. In general, the paper is well written and easy to follow. Two minor suggestions:

Results, first sentence: what is “case 1”, “case 2” etc. I thought there were 8 actors/USPs not 6, is this the number of interviews from each USP? I see from the table this refers to scenarios or scripts. Perhaps this could be better described in the methods?

Did each USP participate in each scenario/case, or did some “specialize” in certain cases, so to speak?

Added: "Two USPs were assigned to each case (some overlapping cases) using scripted commonly occurring PrEP scenarios (Table 1) to seek PrEP services at all 24 facilities." to the Data Collection sub-section.

Reviewers' comments:

Reviewer's Responses to Questions

Comments to the Author

1. Is the manuscript technically sound, and do the data support the conclusions?

Reviewer #1: Yes

2. Has the statistical analysis been performed appropriately and rigorously?

Reviewer #1: N/A

3. Have the authors made all data underlying the findings in their manuscript fully available?

Reviewer #1: Yes

4. Is the manuscript presented in an intelligible fashion and written in standard English?

Reviewer #1: Yes

5. Review Comments to the Author

Reviewer #1: Authors present a large qualitative study of mystery clients who presented as AGYW who were interested to discuss PrEP continuation/prescription. The study was embedded in a large trial involving 24 health facilities. Data were thematically analysed and revealed three themes reflecting on the interactions experienced.

Privacy, attitudes to clients, and 'patient-centered communication' - emerge as main perceived determinants of improved HIV preventive care (i.e. PrEP continuation and initiation).

The paper is well-written reinforces that clinician attitudes impact uptake of prevention products.

The discussion is fairly brief and reiterates the three major findings. I was missing a broader discussion on what it will take to get AGYW to take up daily oral PrEP.

It is clear that improved privacy and client-centred communication is one aspect. There are several other aspects of PrEP programming (e.g. stigma reduction, societal attitudes to sexual activity and PrEP, mental health support, alternative PrEP dispensing approach, role of peer support) that go beyond the mystery clients approach but could be mentioned to contextualise study findings.

The final sentence of the discussion is weak: "Future research is needed that includes a broader sample of non-actor AGYW participants to evaluate of whether provider and health systems interventions to improve these interactions will result in improved uptake and persistence in this priority population."

Especially since authors speak about health systems interventions that have hardly been discussed or put in context.

Small point: how did the sex (gender) of the provider impact interactions?

Thank you for your review and extremely helpful comments. I have added more context to the Discussion and Conclusion sections as stated below.

First paragraph in Discussion; added: "This adds to the broader context in which AGYWs are seeking HIV prevention services where stigma reduction is paramount and changing discriminating attitudes towards HIV prevention and sex are crucial to decreasing HIV transmission. If AGYW can receive better care in the setting they receive care most often (i.e. within the healthcare system among health care providers), it could help with HIV prevention efforts among this population."

Discussion; added ending paragraph: "In the broader context of HIV prevention among AGYW, more is needed: more access points to PrEP, better systems for retention of AGYW to PrEP services, an overall cultural shift toward more understanding and acceptance of AGYW sexual practices and what they do to protect themselves. Retail pharmacies are a promising new addition to PrEP acquisition.33 Purchasing PrEP over-the-counter has shown in recent studies to provide the privacy, good communication, and convenience many AGYWs prefer when accessing PrEP. Peer-support groups are another way to potentially increase PrEP intiation among AGYWs.34 This model utilizes experienced AGYWs to help PrEP-naïve AGYWs take HIV self-tests and then refers them to PrEP services at clinic. Peer-supported PrEP initiation may be another way to bolster PrEP initiation among AGYWs, but they still need a good experience at the clinic once they get there. Our study reinforces the potential impact that health care providers have among AGYWs accessing HIV prevention."

Conclusion; added: "Further, more research is needed to better contextualize and understand the benefits of retail pharmacy-based PrEP services and peer-supported PrEP programs, both aimed at increasing PrEP access points and retaining more clients on PrEP for younger populations."

6. PLOS authors have the option to publish the peer review history of their article (what does this mean?). If published, this will include your full peer review and any attached files.

Do you want your identity to be public for this peer review? For information about this choice, including consent withdrawal, please see our Privacy Policy.

Reviewer #1: Yes: Eduard Sanders

---

## [Editor Report · Decision Letter 1]

6 Aug 2024

“I was given PrEP, but had no privacy”: Mystery shopper perspectives of PrEP counseling for adolescent girls and young women in Kisumu County, Kenya

PONE-D-24-01290R1

Dear Dr. Vera,

We’re pleased to inform you that your manuscript has been judged scientifically suitable for publication and will be formally accepted for publication once it meets all outstanding technical requirements.

Kind regards,

Matt A Price

Academic Editor

PLOS ONE

Additional Editor Comments (optional):

The authors have adequately addressed the reviewer comments
---

## [Editor Report · Acceptance letter]

9 Aug 2024

PONE-D-24-01290R1 

PLOS ONE

Dear Dr. Vera, 

I'm pleased to inform you that your manuscript has been deemed suitable for publication in PLOS ONE. Congratulations! Your manuscript is now being handed over to our production team.

Kind regards, 

on behalf of

Dr. Matt A Price 

Academic Editor

PLOS ONE